# Laser Ranging-Assisted Binocular Visual Sensor Tracking System

**DOI:** 10.3390/s20030688

**Published:** 2020-01-27

**Authors:** Qilong Wang, Yu Zhang, Weichao Shi, Meng Nie

**Affiliations:** 1School of Mechanical and Precision Instrument Engineering, Xi’an University of Technology, Xi’an 710048, China; shiweichao@xaut.edu.cn; 2Beijing Aerospace Times Optical-electronic Technology CO, Ltd., China Aerospace Science and Technology Corp, Beijing 100094, China; zhangyu2006101@126.com; 3School of Mechanical, Electronic and Control Engineering, Beijing Jiaotong University, Beijing 100044, China; nmbc@bjtu.edu.cn

**Keywords:** binocular vision, laser ranging sensor, target tracking, information fusion, time delay

## Abstract

Aimed at improving the low measurement accuracy of the binocular vision sensor along the optical axis in the process of target tracking, we proposed a method for auxiliary correction using a laser-ranging sensor in this paper. In the process of system measurement, limited to the mechanical performance of the two-dimensional turntable, the measurement value of a laser-ranging sensor is lagged. In this paper, the lag information is updated directly to solve the time delay. Moreover, in order to give full play to the advantages of binocular vision sensors and laser-ranging sensors in target tracking, federated filtering is used to improve the information utilization and measurement accuracy and to solve the estimated correlation. The experimental results show that the real-time and measurement accuracy of the laser ranging-assisted binocular visual-tracking system is improved by the direct update algorithm and the federal filtering algorithm. The results of this paper are significant for binocular vision sensors and laser-ranging sensors in engineering applications involving target tracking systems.

## 1. Introduction

Visual measurement has many advantages, such as high accuracy and a non-contact nature. It is widely used in industrial and military applications and daily life [1,2,3]. According to the different number of cameras used in the measurement process, visual measurement techniques can be generally divided into monocular vision, binocular vision, and multieye vision [4,5,6].

Monocular vision has the problem of scale ambiguity. In the process of solving the corresponding points in the monocular visual image, the fundamental matrix or the homography matrix lacks the depth constraint in the decomposition and cannot determine the proportional coefficient [7]. Compared with monocular vision, stereo vision consists of two or more fixed vision sensors that collect target data simultaneously. By establishing the correspondence between views, the scale information can be estimated quickly, and then the depth recovery and the reconstruction and pose estimation of the target can be realized in the Euclidean space. In reference [8], the three-dimensional coordinates of the moving object are obtained by two cameras that measure the spatial constraints of the same spatial point between different image planes. After the 3D a reconstruction of single acquisition, the absolute orientation of the two acquisition point clouds is solved to achieve the target pose estimation and point cloud fusion. Terui. F et al. [9] used binocular vision to estimate the pose of a known semi-cooperative target and verified its effectiveness in a ground test. However, this method is computationally complex and needs to be processed offline. Du. X et al. [10] explored the feasibility of binocular vision to measure non-cooperative targets with the support of European Space Agency, and then Segal et al. [11] extended the method to discrete multi-objective feature recognition and matching. The Bayer and Kalman filtering algorithms are used to improve the pose estimation accuracy of moving targets. Engel et al. [12] extended binocular vision to simultaneous localization and mapping (SLAM) and implemented a fast initialization of the direct method. In addition, the calibration accuracy determines the measurement accuracy for a binocular vision sensor. J. Apolinar Muñoz Rodríguez [13] proposed a technique to perform microscope self-calibration via micro laser line and soft computing algorithms. In this technique, the microscope vision parameters are computed by means of soft computing algorithms based on laser line projection.

In engineering practice, due to the limitations of a single sensor, target measurement technology based on multisensor information fusion with high-precision and high-robustness has become the main method to make up for the shortcoming of a single sensor [14]. Multisensor information fusion (MSIF), which appeared in the late 1970s, is widely used in target tracking and image and signal processing [15,16,17]. MSIF is a method for improving the adaptability and measurement accuracy of the system by integrating the correlation of each sensor in the same scene.

Based on the fusion of vision and inertial measurement unit (IMU), Leutenegger. S et al. [18] integrated the IMU error into the reprojection error cost function to construct a nonlinear error function, and maintained a fixed optimization window through marginalization to ensure the real-time performance of the process. Forster C et al. [19] proposed a preintegration theory that aggregates the inertial measurements of adjacent keyframes into independent motion constraints and fuses them into a factor graph framework. For the fusion of vision and Lidar, Tomic T et al. [20] used a multihead camera and point cloud registration to jointly estimate the motion state of the drone, and they used the map pyramid to improve the efficiency of the algorithm to achieve real-time performance. Zhang J et al. [21] proposed the depth-enhanced monocular odometry (DEMO) method based on positioning and reconstruction of the camera and 3D lidar. Vision sensors can also be used with sonic sensors for automatic parking, and with computed tomography (CT) images for medical diagnosis, with infrared sensors for target tracking, among other applications. For the fusion of vision and laser. J. Apolinar Muñoz Rodríguez and Francisco Carlos Mejía Alanís [22] proposed an accurate technique to perform binocular self-calibration by means of an adaptive genetic algorithm based on a laser line. In this calibration, the genetic algorithm computes the vision parameters through simulated binary crossover (SBX).

In this paper, we propose a method for correcting the distance of binocular vision along the optical axis with a one-dimensional point laser ranging sensor that is installed on a two-dimensional turntable firstly. Limited by the mechanical structure of the two-dimensional turntable in the actual measurement process, the measurement frequency of the laser ranging sensor is lower than that of the binocular vision sensor, resulting in time-delay and redundancy of the measurement information of the entire system. Second, we propose a direct update algorithm based on one-step prediction, which transforms the time-delay information into real-time information, and combines the federal Kalman filtering algorithm to complete the positioning and tracking of spatial targets. Finally, the experimental results show that the system structure and algorithm that we proposed can effectively improve the accuracy and real-time performance of the multisensor system. It should be noted that the parameters of the binocular vision sensor used in this paper are obtained after image processing. Image processing such as camera calibration, distortion correction, etc. is not the research content of this article.

## 2. System Construction

In this paper, the experimental platform of the target tracking system is mainly composed of two parts: the measurement system and the target system. The measurement system is a typical heterogeneous sensor system that is composed of a binocular vision sensor system and a laser-ranging sensor system. The laser-ranging sensor is mounted on a two-dimensional (2D) turntable. The target system consists of a model that moves via a two-dimensional moving slide-table. The space schematic of the target tracking system is shown in Figure 1; the physical experimental system is shown in Figure 2.

As shown in Figure 2, $O_{1}$ and $O_{2}$ are two cameras of the binocular vision sensor. A is the point of the laser ranging sensor that is fixed in the center of the two-axis turntable. C is the point of the target. According to the spatial location relation of each sensor and target, the coordinate systems $O_{1}\mathrm{XYZ}$ and ${\mathrm{AX}}^{\prime }Y^{\prime }Z^{\prime }$ are established.

The coordinate system $O_{1}\mathrm{XYZ}$: camera $O_{1}$ is the ordinate origin. The extension to the other camera $O_{2}$ is in the X-axis direction and the Y-axis is vertical to $O_{1}O_{2}$. The right-hand rule is used to determine that the direction along the optical axis is the Z-axis. The positive direction of the coordinate axes is shown in Figure 2. The coordinates of target *C* are ($x_{c},y_{c},z_{c}$) in coordinate system $O_{1}\mathrm{XYZ}$.

The coordinate system ${\mathrm{AX}}^{\prime }Y^{\prime }Z^{\prime }$: the point A is the ordinate origin. The directions of the X-axis, Y-axis and Z-axis are the same as in the coordinate system $O_{1}\mathrm{XYZ}$. The coordinates of target *C* and camera $O_{1}$ are ($x_{c}{}^{\prime },y_{c}{}^{\prime },z_{c}{}^{\prime }$) and ($x_{1},y_{1},z_{1}$) in the coordinate system ${\mathrm{AX}}^{\prime }Y^{\prime }Z^{\prime }$, respectively.

The projection of point C on surface $X^{\prime }{\mathrm{AY}}^{\prime }$ is $C_{0}$. The angle between $AC_{0}$ and the $Y^{\prime }$ axis is α. The angle between AC and surface $X^{\prime }{\mathrm{AZ}}^{\prime }$ is β.

The steps for obtaining the spatial location of the target are as follows:

(1) Coordinate system $O_{1}\mathrm{XYZ}$ is set as the reference coordinate system. The binocular vision sensor system first acquires the spatial location of the target, and then transmits the measurement data to the central control system through the visual control computer.

(2) The target space information acquired by the binocular vision sensor is the coordinates in the coordinate system $O_{1}\mathrm{XYZ}$. Coordinate transformation to coordinate system ${\mathrm{AX}}^{\prime }Y^{\prime }Z^{\prime }$ is required to obtain the pitch and yaw angles by which the turntable must be rotated.

The coordinates ($x_{c}{}^{\prime },y_{c}{}^{\prime },z_{c}{}^{\prime }$) are given as follows:(1)$$\begin{array}{l} \left[x_{c}^{\prime },y_{c}^{\prime },z_{c}^{\prime }]=[x_{c},y_{c},z_{c}]\times \left[\begin{array}{ccc} 1 & 0 & 0\\ 0 & \cos m & -\sin m\\ 0 & \sin m & \cos m \end{array}\right]\times \left[\begin{array}{ccc} \cos n & 0 & \sin n\\ 0 & 1 & 0\\ -\sin n & 0 & \cos n \end{array}\right]\times \left[\begin{array}{ccc} \cos k & -\sin k & 0\\ \sin k & \cos k & 0\\ 0 & 0 & 1 \end{array}\right]+[x_{1},y_{1},z_{1}\right]\\ =[x_{c}^{},y_{c}^{},z_{c}^{}]\times \mathrm{Rot}(x,m)\times \mathrm{Rot}(y,n)\times \mathrm{Rot}(z,k)+[x_{1},y_{1},z_{1}] \end{array}$$
where Rot(x,m) is the rotation matrix around the $X^{\prime }$ axis. m is the angle of rotation which is a constant value and is obtained from the initial calibration. Rot(y,n) and Rot(z,k) are the same.

It is obtained from the spatial geometry of Figure 2:(2)$$\alpha =\arctan(\frac{x_{c}^{\prime }}{z_{c}^{\prime }}),$$
(3)$$\beta =\arctan(\frac{\left|y_{c}^{\prime }\right|}{\sqrt{x_{c}^{\prime }{}^{2}+z_{c}^{\prime }{}^{2}}}),$$

(3) The yaw angle α and pitch angle β of the 2D turntable are transferred to the laser control computer. Then, after the 2D turntable is adjusted to the designated position, the laser ranging sensor is controlled to shoot towards target *C* through the laser control computer and the distance value AC=l is returned to the central control system.

(4) The measurement values l of the laser ranging sensor are used to correct the measured value of the binocular vision sensor along the optical axis direction (Z axis). Because of the high accuracy of the binocular vision sensor along the vertical optical axis, the coordinates on the X axis and Y axis can be used as the measured values.

The Z coordinate of point C in the coordinate system ${\mathrm{AX}}^{\prime }Y^{\prime }Z^{\prime }$ is obtained as follows:(4)$$z_{c}^{\prime }=l\times \cos\alpha \times \cos\beta ,$$

By the coordinate transformation, the $Z^{\prime }$ coordinate of point C is obtained:(5)$$z_{c}^{\prime }=z_{c}\times \mathrm{Rot}(x,m)\times \mathrm{Rot}(y,n)\times \mathrm{Rot}(z,k)+z_{1},$$

Therefore, the new *Z* coordinate value of the *C* point $z_{nc}$ in the coordinate system $O_{1}\mathrm{XYZ}$ after the correction by the laser ranging sensor can be solved:(6)$$z_{nc}=(l\times \cos\alpha \times \cos\beta )/[\mathrm{Rot}(x,m)\times \mathrm{Rot}(y,n)\times \mathrm{Rot}(z,k)],$$

Through error analysis and the error transfer formula, the error of $z_{nc}$ can be obtained:(7)$$\Delta z_{nc}=\frac{\cos\alpha \cos\beta |\Delta l|+l\sin\alpha \cos\beta |\Delta \alpha |+l\cos\alpha \sin\beta |\Delta \beta |}{\mathrm{Rot}(x,m)\times \mathrm{Rot}(y,n)\times \mathrm{Rot}(z,k)},$$
where, Δl is the measurement error of laser-ranging sensor, Δα is the error of the yaw angle, Δβ is the error of the pitch angle.

In practice, because the position calibration error between the binocular vision sensor and the laser-ranging sensor is quite small, the primary error is concentrated in the above three errors. The measurement error of the laser-ranging sensor is determined by its own performance. The angle error includes the calculation error caused by binocular vision sensor and the rotation error of the 2D turntable. Therefore, the spatial position error of the target corrected by the laser-ranging sensor is affected by the measurement error of the binocular vision sensor.

## 3. Problem Description

According to the calculation process of the coordinate $z_{c}$, the information flow of the target tracking is shown in Figure 3.

The measured value $z_{k-1}^{c}$ is obtained by the binocular vision sensor at time $t_{k-1}$. Then, the yaw angle α and pitch angle β are calculated by coordinate transformation and transferred to the 2D turntable controller. After the turntable is rotated to the corresponding angle, the laser-ranging sensor is tested and the corrected coordinate value $z_{k-1}^{nc}$ at time $t_{k}$ is calculated. Since the frequency of mechanical rotation of the 2D turntable is much lower than the measurement frequency of the binocular vision sensor, the measured value $z_{k}^{c}$ is obtained by the binocular vision sensor at time $t_{k}$.

From the process of constructing the system and the acquisition of the target space position, the measurement information of the binocular vision sensor and that of the laser-ranging sensor are related, and the laser-ranging sensor system has constant time delay.

Consider the following multiple sensors system with observing time-delay:(8)$$x_{k}=Fx_{k-1}+G_{}w_{k,k-1},$$
(9)$$z_{k}^{i}=Hx_{k-d_{i}}+v_{k}^{i},$$
where, $1\leq d_{1}\leq d_{2}\leq \cdots \leq d_{N}\leq d_{i}$
(i=1,2,⋯,N), in which $d_{i}$ is the observation delay of the *i*th local sensor; $x_{k}\in R^{n\times 1}$ is the state vector of system at time $t_{k}$, $F\in R^{n\times n}$ is the n×n dimensional state-transitional matrix. $G\in R^{n\times h}$ is the n×h dimensional noise input matrix. $z_{k}^{i}\in R^{m\times 1}\begin{array}{cc} , & i \end{array}=1\cdots L$ is the m-dimensional measured vector of the *i*th sensor, and L is the number of sensors. $H^{i}\in R^{m\times n}\begin{array}{cc} , & i \end{array}=1\cdots L$ is the measured matrix of the *i*th sensor. $w_{k,k-1}\in R^{h\times 1}$ is h-dimensional process noise vector of *i*th sensor. $v_{k}^{i}\in R^{m\times 1}\begin{array}{cc} , & i \end{array}=1\cdots L$ is the observation noise of the *i*th sensor.

In the real-time sensor system, at time $t=t_{k}$, the following can be obtained:(10)$$\left\{ \begin{array}{l} {\hat{x}}_{k|k}=E^{*}[x_{k}|Z^{k}]\\ P_{k|k}=\mathrm{cov}[x_{k}|Z^{k}] \end{array}\right.,$$
where $Z^{k}$ is the measurement set $Z^{k}={\left\{z_{i}^{N}\right\}}_{i=1}^{k}$ of the Nth sensor at time $t_{k}$.

Assume that the lag time of the time-delay sensor (laser ranging sensor) is $t_{k-d}$. There are real-time measurement $z_{k}^{i}$ (binocular vision sensor) and time-delay measurement $z_{k-d}^{j}$ (laser ranging sensor) in the fusion center at time $t_{k}$. We need to use earlier measurements of the time-delay sensor to update the estimation ${\hat{x}}_{k|k}$:(11)$$\left\{ \begin{array}{l} {\hat{x}}_{k|k,d}=E^{*}[x_{k}|Z^{k},z_{d}]\\ P_{k|k,d}=\mathrm{cov}[x_{k}|Z^{k},z_{d}] \end{array}\right.,$$

Moreover, the estimated value ${\hat{x}}_{k|k-d}$ of the time-delay sensor system at time $t_{k}$ is considered to be the real-time measured value $z_{k}^{j}$ at time $t_{k}$. The measured value of all sensors at time $t_{k}$ are $\left\{z_{k}^{1},z_{k}^{2},\cdots ,z_{k}^{i},\cdots ,z_{k}^{j}\right\}$(1≤i≤j≤N), where the estimates of $z_{k}^{i}$ and $z_{k}^{j}$ are relevant. To obtain more accurate space coordinates of the target, we need to resolve the above correlation problem and the optimal fusion estimation of the target motion state $x_{k}$ in the fusion center.

## 4. Time-Delay Information Update and Fusion

### 4.1. Processing of Measurement Constant Time Lag

Based on the stochastic linear time-invariant discrete system and the iterated state equation, we can obtain the following:(12)$$x_{k+1}=F^{d_{i}}x_{k+1-d_{i}}+\sum_{j=1}^{d_{i}}F^{j-1}Gw_{k+1-j},$$

Namely,
(13)$$x_{k-d_{i}}=F^{-d_{i}}x_{k}-\sum_{j=1}^{d_{i}}F^{j-1-d_{i}}Gw_{k-j},$$

Inserting the above formula into the observation equation,
(14)$$z_{k}^{i}=H^{i}F^{-d_{i}}x_{k}-\sum_{j=1}^{d_{i}}H^{i}F^{j-1-d_{i}}Gw_{k-j}+v_{k}^{i},$$

Assuming ${\bar{H}}^{i}=H_{}^{i}F^{-d_{i}}$,$\eta_{k}^{i}=v_{k}^{i}-\sum_{j=1}^{d_{i}}H^{i}F^{j-1-d_{i}}Gw_{k-j}=v_{k}^{i}-\sum_{j=1}^{d_{i}}{\bar{H}}^{i}F^{j-1}Gw_{k-j}$,

Then the system observation equation is transformed into:(15)$$z_{k}^{i}={\bar{H}}^{i}x_{k}+\eta_{k}^{i},$$

Combined with Equation (15), the time-delay subsensor system has the following optimal Kalman filter and one-step predictor,
(16)$${\hat{x}}_{k+1|k+1}^{i}={\hat{x}}_{k+1|k}^{i}+K_{k+1}^{i}\varepsilon_{k+1}^{i},$$
(17)$${\hat{x}}_{k+1|k}^{i}=F{\hat{x}}_{k|k}^{i},$$
(18)$$\varepsilon_{k+1}^{i}=z_{k+1}^{i}-{\hat{z}}_{k+1|k}^{i}=z_{k+1}^{i}-{\bar{H}}^{i}{\hat{x}}_{k+1|k}^{i},$$
(19)$$K_{k+1}^{i}=(P_{k+1|k}^{i}{\left({\bar{H}}^{i}\right)}^{T}-\sum_{j=1}^{d_{i}}F^{j-1}GQG^{T}{\left(F^{j-1}\right)}^{T}{\left({\bar{H}}^{i}\right)}^{T})P_{\varepsilon }^{i},$$
(20)$$P_{\varepsilon }^{i}={\bar{H}}^{i}P_{k+1|k}^{i}{\left({\bar{H}}^{i}\right)}^{T}+R^{ii}-\sum_{j=1}^{d_{i}}{\bar{H}}^{i}F^{j-1}GQG^{T}{\left(F^{j-1}\right)}^{T}{\left({\bar{H}}^{i}\right)}^{T},$$
(21)$$P_{k+1|k}^{i}=FP_{k|k}^{i}F^{T}+GQG^{T},$$
(22)$$P_{{}_{k+1|k+1}}^{i}=P_{k+1|k}^{i}-K_{k+1}^{i}P_{\varepsilon }^{i}{\left(K_{k+1}^{i}\right)}^{T},$$
where, $\varepsilon_{k+1}^{i}$ is the innovation. $P_{\varepsilon }^{i}=E[\varepsilon_{k+1}^{i}{\left(\varepsilon_{k+1}^{i}\right)}^{T}]$ is the innovation variance. $K_{k+1}^{i}$ is the filter gain. $P_{{}_{k+1|k+1}}^{i}$ is the filtering error variance matrix. $P_{k+1|k}^{i}$ is the one step prediction error variance matrix.

### 4.2. Information Fusion

To solve the estimation correlation between two sensor systems and further improve the accuracy of measurement, the federal Kalman filtering algorithm is used for subsequent processing, which includes information distribution, time updating, measurement updating, and estimation fusion.

(1) Information distribution

The main filter only updates the timing and dose not measure. The process information of the system is shared among the subfilters and the main filters according to the principle of information distribution.
(23)$$\left\{ \begin{array}{l} P_{k-1}^{i}=\beta_{i}^{-1}P_{k-1}\\ Q_{k-1}^{i}=\beta_{i}^{-1}Q_{k-1}\\ x_{k-1}^{i}=X_{k-1} \end{array}\right..$$

According to the Law of Information Conservation, $\sum_{i=0}^{n}\beta_{i}=1$.

(2) Time updating

The covariance of the system state and estimation error is transferred according to the system transfer matrix, which is performed independently for the sub-filter and the main filter.
(24)$${\hat{x}}_{k|k-1}^{i}=F_{k|k-1}^{i}{\hat{x}}_{k-1}^{i},$$
(25)$$P_{k|k-1}^{i}=F_{k|k-1}^{i}P_{k-1}^{i}{\left(F_{k|k-1}^{i}\right)}^{T}+G_{k-1}^{i}Q_{k-1}^{i}{\left(G_{k-1}^{i}\right)}^{T}.$$

(3) Measurement updating

The system state and estimated error covariance are updated using the new measurement information. Since the main filter performs no measurements, the measurement updating is only performed in the subfilter.
(26)$$K_{k}^{i}=P_{k|k-1}^{i}{\left(H_{k}^{i}\right)}^{T}{\left(R_{k}^{i}\right)}^{-1},$$
(27)$${\hat{x}}_{k}^{i}={\hat{x}}_{k|k-1}^{i}+K_{k}^{i}(z_{k}^{i}-H_{k}^{i}{\hat{x}}_{k|k-1}^{i}),$$
(28)$${\left(P_{k}^{i}\right)}^{-1}={\left(P_{k|k-1}^{i}\right)}^{-1}+{\left(H_{k}^{i}\right)}^{T}{\left(R_{k}^{i}\right)}^{-1}H_{k}^{i},$$

(4) Estimation fusion
(29)$${\hat{X}}_{g}=P_{g}{\sum \left(P_{k}^{i}\right)}^{-1}{\hat{x}}_{k}^{i},$$
(30)$$P_{g}={\left(\sum {\left(P_{k}^{i}\right)}^{-1}\right)}^{-1}.$$

In above steps, information distribution is a key part of federated filtering, which is an important feature that distinguishes it from other decentralized filtering methods. The coefficient of the information distribution determines the accuracy of the final fusion result.

According to the estimated error covariance,
(31)$$P_{i}=E[(X_{i}-{\overset{⌢}{X}}_{g}){\left(X_{i}-{\overset{⌢}{X}}_{g}\right)}^{T}].$$

It can be seen that P describes the estimation accuracy of X, and the smaller the P, the higher the estimation accuracy of X.

Considering the use of a globally optimal solution to reset the filter values and error variance matrices in the next filtering step, the influence of the information distribution coefficients on global estimates is discussed:

By inserting Equation (30) into Equation (23), the following is obtained:(32)$$P_{k+1}^{i}=\beta_{i}^{-1}P_{g}.$$

Equations (30) and (32) are substituted into Equation (25) to obtain:(33)$${\hat{X}}_{k+1|k}^{i}=F_{k+1|k}^{i}{\hat{X}}_{g,k}=F_{k+1|k}^{i}P_{g}\sum (\beta_{i}{}^{-1}P_{k}^{i}{)}^{-1}{\hat{x}}_{k}^{i}$$
where ${\hat{x}}_{k}^{i}=F_{k|k-1}^{i}{\hat{x}}_{k-1}^{i}+K_{k}^{i}(z_{k}^{i}-H_{k}^{i}{\hat{x}}_{k|k-1}^{i})$.

P and Q have the same dimension in general. It is given by the following:(34)$$P_{i}=E[(X_{i}-{\overset{⌢}{X}}_{g}){\left(X_{i}-{\overset{⌢}{X}}_{g}\right)}^{T}]$$
(35)$$\begin{array}{ll} P_{k+1|k}^{i} & =F_{k+1|k}^{i}P_{k}^{i}{\left(F_{k+1|k}^{i}\right)}^{T}+G_{k}^{i}Q_{k}^{i}{\left(G_{k}^{i}\right)}^{T}=F_{k+1|k}^{i}\beta_{i}^{-1}P_{g,k}^{}{\left(F_{k+1|k}^{i}\right)}^{T}+G_{k}^{i}\beta_{i}^{-1}Q_{g,k}^{}{\left(G_{k}^{i}\right)}^{T}\\ & =\beta_{i}^{-1}(F_{k+1|k}^{i}P_{g,k}^{}{\left(F_{k+1|k}^{i}\right)}^{T}+G_{k}^{i}Q_{g,k}^{}{\left(G_{k}^{i}\right)}^{T})=\beta_{i}^{-1}P_{k+1|k}^{{}^{\prime }i} \end{array}$$
where $P_{k+1|k}^{{}^{\prime }i}≜F_{k+1|k}^{i}P_{g,k}^{}{\left(F_{k+1|k}^{i}\right)}^{T}+G_{k}^{i}Q_{g,k}^{}{\left(G_{k}^{i}\right)}^{T}$.

Taking the inverse of the Equation (34) on both sides, it is the one-step predictive state information matrix of local filter and global filter.
(36)$${\left(P_{k+1|k}^{i}\right)}^{-1}=\beta_{i}^{}{\left(P_{k+1|k}^{{}^{\prime }i}\right)}^{-1}$$

Taking the trace of both sides, we can obtain:(37)$$\beta_{i}=\frac{tr[{\left(P_{k+1|k}^{i}\right)}^{-1}]}{tr[{\left(P_{k+1|k}^{{}^{\prime }i}\right)}^{-1}]}$$

$\beta_{i}$ is inversely proportional to the estimated error covariance. When the estimation covariance is larger, the estimation quality is poorer, the subfilter accuracy is lower, and the information distribution coefficient is smaller.

## 5. System Experiment and Analysis

The target-tracking system of binocular vision laser-ranging sensor is shown in Figure 1. The target is fixed on a sliding platform and performs linear motion in space. At present, only the movement of the target in the direction of the optical axis (Z axis) is studied, and the measured values in the experimental results only represent the measured values in the Z axis direction.

Through error analysis of the laser-ranging sensor, the parameters of the system are brought into the error transfer function. The measurement error of the laser-ranging sensor is ±1.5mm, the rotation error of the two-dimensional turntable is $\pm 0.02^{\circ }$, the measurement error of the binocular vision X and Y directions is ±3mm, and the Z-axis measurement error is ±40mm. After several calculations, the final average measurement error of the laser-ranging sensor is ±22.3mm, which proves that the laser-ranging sensor improves the measurement accuracy of binocular vision along the optical axis.

As shown in Figure 4, the measurement value of the laser-ranging sensor is directly predicted, and the estimated value of the lag information is used as real-time information for subsequent calculations. It can be seen from the simulation results at the time of t (25) to t (40) that the lag information is improved after the direct update algorithm, and the estimated value has errors due to the influence of noise, but the change trend is basically consistent with the original data. At the same time, when observing the whole curve, the target position changes slowly at the beginning stage, and the slope of the curve gradually increases with the passage of time, and then basically remains unchanged from t (38). This is because there is acceleration at the beginning of the sliding platform where the target is located. After reaching the specified speed, the target enters the stage of uniform motion, and the slope does not change. The target moves for a long time, and the curve of the subsequent deceleration phase is not drawn in Figure 4.

Then, the information fusion algorithm based on the federated Kalman filter is verified by experiments. A comparison between the measurement results of a single sensor and the fusion results is shown in Figure 5.

It can be clearly observed in Figure 5 that compared with the binocular vision sensor with larger error, the curve of the fusion result is smoother, and the accuracy is improved.

Compared with the measured value of the laser-ranging sensor, the fusion result can not be seen directly, so the mean squared error (MSE) between each measurement result and the calibrated true value of the target is calculated. The MSE results in Figure 6 indicate that the accuracy of the target position after fusion is improved compared to a single sensor, and the error of the result after fusion is the smallest.

Figure 7 shows the change curve of the binocular vision sensor’s information distribution coefficient. At the beginning of the measurement, the information distribution coefficient changes rapidly from 0.5 in 5 S. After 15 s, it tends to be stable and its value is 0.37. Because the measurement error of the binocular vision sensor along the optical axis is large, its information distribution coefficient is small, while the laser-ranging sensor has a large information distribution coefficient, which is consistent with the conclusions obtained in the paper.

## 6. Conclusions

Through theoretical calculations and experimental verifications, the accuracy of binocular vision along the optical axis is improved in this paper. First, regarding the system structure, we propose to use a one-dimensional point laser-ranging sensor to correct the measurement value of the binocular vision sensor along the optical axis. Second, regarding the measurement process, we found that it is limited by the performance of the two-dimensional turntable, the system has time delay. To improve the utilization and real-time performance of the information of the multisensor measurement system, an optimal information fusion algorithm for the multisensor target tracking system characterized by estimation correlation and constant time delay is studied. We propose a method to separate the complex multisensor environment, which first uses the one-time prediction of the constant delay information as the real-time information at the current moment to solve the delay problem, and then uses the federal Kalman filter to address the estimation correlation. Finally, the experimental results verify the validity and accuracy of this method.

Although the research in this article provides a basis for the study of the experimental system in actual environments, the current experimental environment is relatively simple, and the experimental parameters and errors are under control. When the distance between the target and sensor system increases, the visual error will increase, and the laser spot will become larger, which will cause the errors of the laser-ranging sensor to increase. There are still many improvements to be studied.

## Figures and Tables

**Figure 1 sensors-20-00688-f001:**
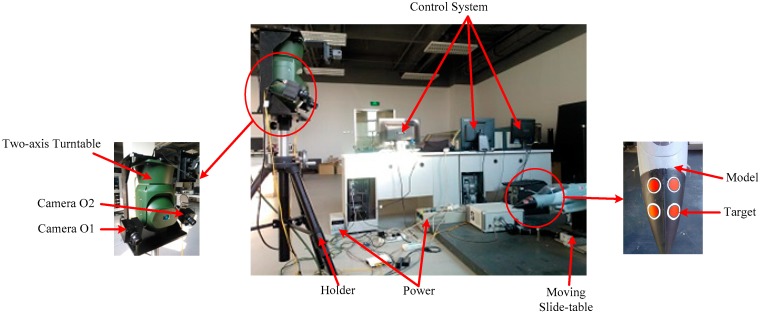
The physical experimental system. The target to be tracked is a red round sticker attached to the model.

**Figure 2 sensors-20-00688-f002:**
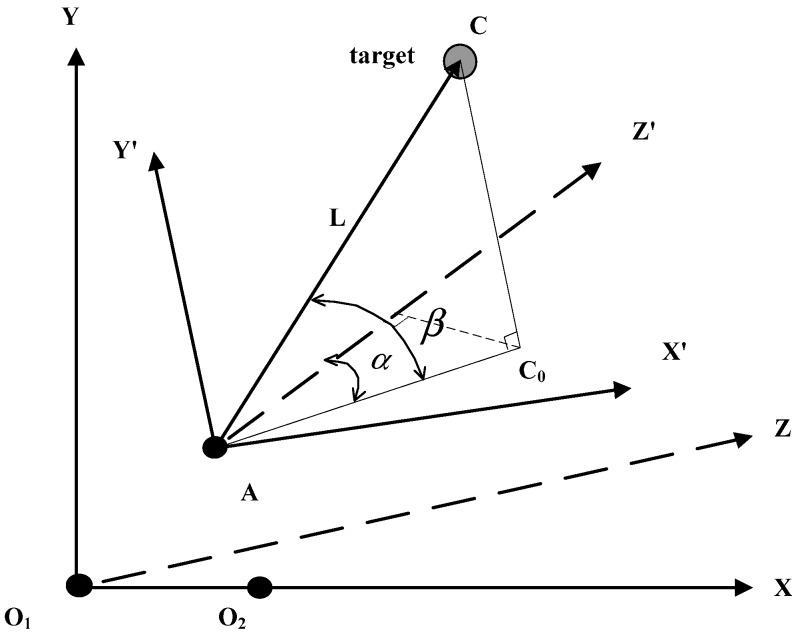
The space geometry relation of a multisensor system.

**Figure 3 sensors-20-00688-f003:**
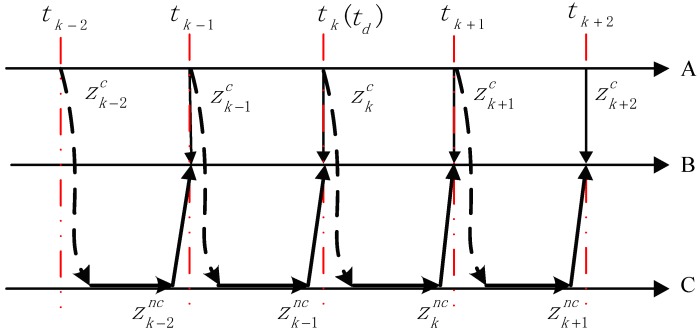
The measurement message passing of system: (**A**): the measurement time of binocular vision; (**B**): the measurement time of fusion; (**C**): the measurement of laser ranging sensor.

**Figure 4 sensors-20-00688-f004:**
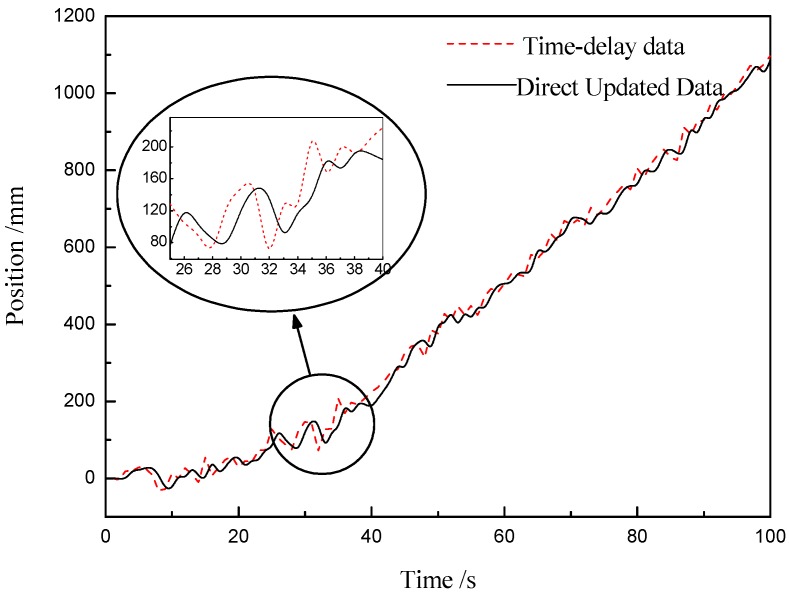
Direct update of time-delay data.

**Figure 5 sensors-20-00688-f005:**
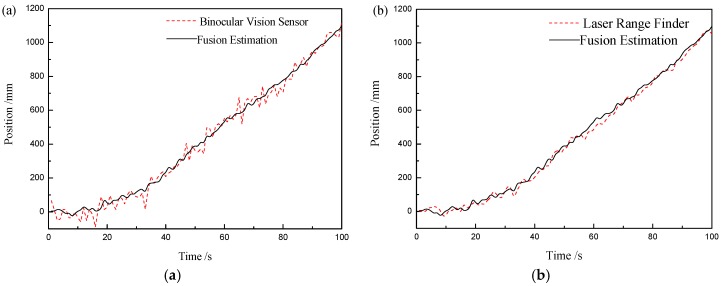
The tracking results for two sensors and fusion. (**a**) Tracking via the binocular vision sensor; (**b**) tracking via the laser range sensor.

**Figure 6 sensors-20-00688-f006:**
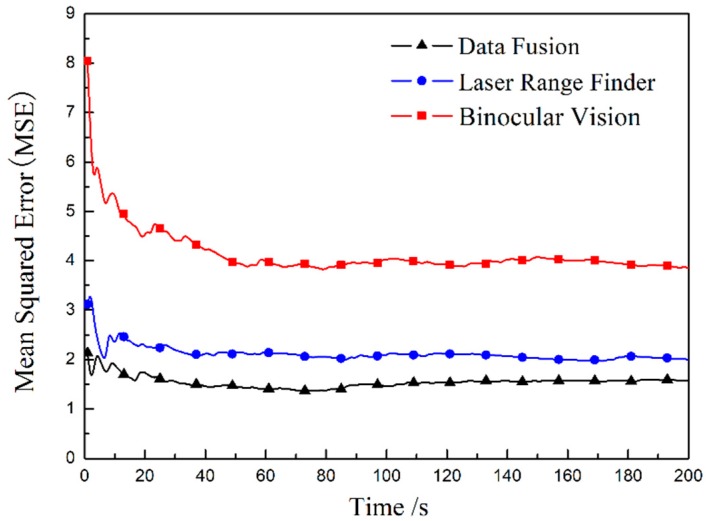
The mean squared error (MSE) of sensors and fusion results.

**Figure 7 sensors-20-00688-f007:**
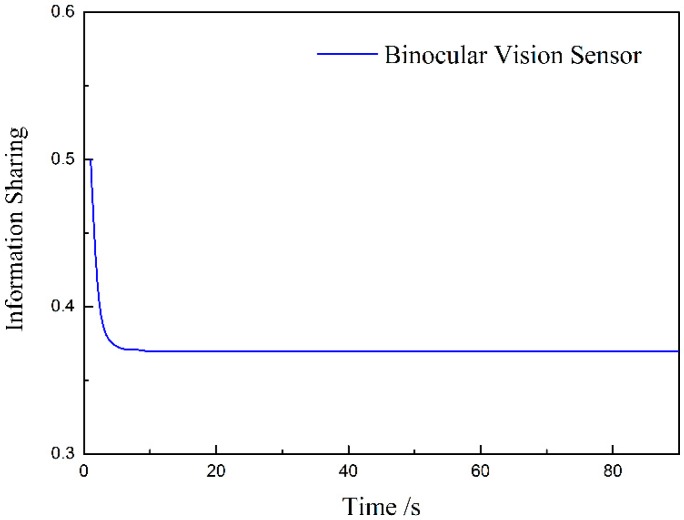
The binocular vision information distribution.

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
