# Peer review of "Laser Ranging-Assisted Binocular Visual Sensor Tracking System"

_sensors, 2020, doi:10.3390/s20030688_

Round 1
Reviewer 1 Report
This article is interesting and the authors show a real scientific background but however some points are a bit weak:
- It seems to me that the authors should pay attention to the use of the word "accuracy". The distinction between precision and accuracy should be made in the article especially when it comes to the evaluation of a calibration method.
ex.: line 603: experimentation is not the only way to assess accuracy!!! ! The term stability is not the right one.
-Sometimes mathematical developments may have no place in this article and have the effect of unnecessarily 'filling' the article. For example the developments on obtaining Euler angles from rotation matrices, (equation (10) and others are now well known, just refer to the right bibliography.
Similarly the developments on image rectification may be useless, the authors have quoted Mr. Andrea Fusiello, others may be also relevant but the presentation of image rectification as the calculation of Euler angles is not the main subject of the article.
-reference to DLT line 46 is more about epistemology than the state of the art.
-Regarding radial distortion, the authors could perhaps have also used the k4 coefficient, this approach and even beyond is more and more used when looking for precision.
It seems to me that personally I would not have proceeded this way: it is sufficient for the evaluation of the proposed method to use a three-dimensional network of targets measured for example with a total station and to use the calibration parameters to calculate the residuals of reprojection on the images after an orientation calculation done correctly with an adjustment bundle. This effectively requires the use of professional photogrammetry software or developments integrating a quality adjustment bundle, for example the one developed by Lourakis.
Finally, the main weakness of the paper is that the extrinsic parameters are calculated without the base. Indeed the baseline which according to the authors can be measured (line 440) should be calculated and this implies the knowledge of an object or points in the scene. Because as the authors say in line 33 the baseline can be deformed ( dilate ) for example due to a temperature variation.
Authors should specify more clearly that the calibration is incomplete, that baseline determination is not part of the process, and also note that their approach is not as innovative as that.
Author Response
Please see the attachment, thank you.

Reviewer 2 Report
Manuscript number: sensors-691652
Title: Laser ranging-assisted binocular visual sensor tracking system
Authors: Qilong Wang, Yu Zhang, Weichao Shi and Meng Nie
The Laser ranging-assisted binocular visual sensor is an interesting task. I agree with the concepts explained in the paper. The mathematical fundamentals are well described. However, the paper lacks some matters to establish the powerful of the proposed method. Therefore, some comments should be included in the manuscript.
1.- Firstly, the binocular system based on laser has been to determine 3D surface measurements [1]. What is the approach respect to the proposed technique? Comments about these matters should be included.
2.- The calibration accuracy determines the measurement accuracy. And the calibration accuracy is determined by the optimization method [2], whose accuracy is given as a pixel fraction. However, the manuscript does not describe the calibration accuracy. Moreover, the optimization procedure is not described. Comments about these matters should be included.
3.- The main parameter in a vision system is the image distortion coefficients. These coefficients are not described in the paper. Comments about these matters should be included.
4.- The most important procedure to determine the powerful of the system is to determine three-dimensional coordinates. Comments about these matters should be included.
5.- The accuracy is determined in the paper via MSE, to provide better accuracy information the accuracy should be pointed via relative error in terms of percentage.
6.- Finally, Some typographic errors should be corrected in the manuscript.
[1] J. of Modern optics, Vol.63 No.13, p.1219-1232, (2016).
[2] OLEN, Vol. 105, p.75–85, (2018).
